# *Gymnodinium catenatum* Paralytic Shellfish Toxin Production and Photobiological Responses under Marine Heat Waves

**DOI:** 10.3390/toxins15020157

**Published:** 2023-02-14

**Authors:** Vanessa M. Lopes, Mélanie Court, Martim Costa Seco, Francisco O. Borges, Bernardo Vicente, Sandra Lage, Ana Catarina Braga, Bernardo Duarte, Catarina Frazão Santos, Ana Amorim, Pedro Reis Costa, Rui Rosa

**Affiliations:** 1MARE—Marine and Environmental Sciences Centre & ARNET—Aquatic Research Network Associate Laboratory, Laboratório Marítimo da Guia, Faculdade de Ciências da Universidade de Lisboa, Avenida Nossa Senhora do Cabo 939, 2750-374 Cascais, Portugal; 2Departamento de Biologia Vegetal, Faculdade de Ciências da Universidade de Lisboa, 1749-016 Lisboa, Portugal; 3MARE—Marine and Environmental Sciences Centre & ARNET—Aquatic Research Network Associate Laboratory, Faculdade de Ciências da Universidade de Lisboa, Campo Grande, 1749-016 Lisbon, Portugal; 4CCMAR—Centre of Marine Sciences, Campus de Gambelas, University of Algarve, 8005-139 Faro, Portugal; 5IPMA—Portuguese Institute for the Sea and Atmosphere, 1749-077 Lisboa, Portugal; 6S2AQUA—Collaborative Laboratory, Association for a Sustainable and Smart Aquaculture, 8700-194 Olhão, Portugal; 7Departamento de Biologia Animal, Faculdade de Ciências da Universidade de Lisboa, 1749-016 Lisboa, Portugal

**Keywords:** dinoflagellate, climate change, marine heatwaves, PSP, saxitoxin

## Abstract

Marine heatwaves (MHWs) have doubled in frequency since the 1980s and are projected to be exacerbated during this century. MHWs have been shown to trigger harmful algal blooms (HABs), with severe consequences to marine life and human populations. Within this context, this study aims to understand, for the first time, how MHWs impact key biological and toxicological parameters of the paralytic shellfish toxin (PST) producer *Gymnodinium catenatum*, a dinoflagellate inhabiting temperate and tropical coastal waters. Two MHW were simulated—category I (i.e., peak: 19.9 °C) and category IV (i.e., peak: 24.1 °C)—relative to the estimated baseline in the western coast of Portugal (18.5 °C). No significant changes in abundance, size, and photosynthetic efficiency were observed among treatments. On the other hand, chain-formation was significantly reduced under category IV MHW, as was PSP toxicity and production of some PST compounds. Overall, this suggests that *G. catenatum* may have a high tolerance to MHWs. Nevertheless, some sublethal effects may have occurred since chain-formation was affected, suggesting that these growth conditions may be sub-optimal for this population. Our study suggests that the increase in frequency, intensity, and duration of MHWs may lead to reduced severity of *G. catenatum* blooms.

## 1. Introduction

Changes to the ocean environment induced by climate change by the end of the century are bound to affect ocean systems and biological dynamics [1]. When studying the effects of climate change on marine biota there has been a strong emphasis on ocean warming [2] compared to other climate-related drivers of change (e.g., acidification and deoxygenation). Indeed, long-term temperature changes present a severe challenge for marine species. However, it is expected that short-term extreme events, such as marine heatwaves (MHW), will lead to severe bottlenecks in population survival [1]. This situation is especially worrisome for species already near their thermal tolerance limits. MHWs can be characterised as events where anomalously high sea surface temperatures (SSTs) are registered, exceeding the 90th percentile of daily SSTs, potentially lasting from five days to several months, and spanning over thousands of square kilometres [3,4]. Hobday et al. [4] categorised MHWs according to their intensity, as the “local difference between the climatological mean and the climatological 90th percentile”, in the following categories: moderate (i.e., category I: 1–2 times the local difference), strong (i.e., category II: 2–3 times), severe (i.e., category III: 3–4 times), and extreme (i.e., category IV: 4 times and above).

The frequency of these extreme events has nearly doubled since the 1980s, and it is expected to continue increasing also in intensity and duration [5]. Alongside the increasing frequency and severity of MHWs, there is an increased potential for more frequent outbreaks of algal blooms. Indeed, temperature fluctuations may be one of the main drivers in phytoplankton species composition and abundance [6,7]. Typically, with increasing temperatures, phytoplanktonic species tend to have higher growth rates until a species-specific temperature threshold is reached [8]. These events can become problematic if the species undergoing the sudden overgrowth produce toxins and can originate harmful algal blooms (HABs). HABs are usually associated with severe negative impacts on marine ecosystems as well as impacts on the human economy and populations [9]. In fact, one remarkable event developed in late 2013 in the northeast Pacific Ocean, dubbed “the Blob”. During this event, which lasted from 2013 until 2015, SSTs reached over three standard deviations above historical data [10]. The Blob triggered a large bloom of toxin-producing diatoms (*Pseudo-nitzschia australis*) that lead to massive fish kills and a great increase in toxin production [11], yielding the highest toxin concentration registered on the North American coast [10]. Besides the impacts on marine communities, this MHW caused widespread fishery closures, leading to overwhelming economic impacts [12] Another, more recent extreme MHW off the coast of Japan in 2021 led to an extensive HAB, with the toxic dinoflagellate *Karenia selliformis* being the most abundant in phytoplankton samples [13,14].

The toxic dinoflagellate *Gymnodinium catenatum* Graham is a species that inhabits temperate to tropical waters around the globe [15,16]. This species is a chain-forming, athecate dinoflagellate which produces paralytic shellfish toxins (PSTs), a group of compounds composed by saxitoxin and its derivatives. PSTs are known to elicit a wide range of impacts, ranging from recoverable effects to events of mass mortality in marine communities (for review see [17]). Throughout the years, several studies have investigated the effects of temperature on the growth rates, life cycle transitions, and toxin profiles of *G. catenatum* [16,18,19]. They were carried out using isolates from different areas, with the objective of using toxin profiles as biochemical markers to distinguish populations and provide knowledge on the thermal range that this species can inhabit. To date, no study has investigated the impact of predicted temperature increases related to climate change or MHW exposure.

Understanding the potential impacts of short-term extreme events such as MHWs is of vital importance, given the reduced opportunity for species adaptation and the particularly severe repercussions that these events can have on marine ecosystems [3,20] and human populations. The hypothesis postulated here is that MHWs will affect *G. catenatum* growth and toxin production. Therefore, the aim of the present study was to investigate, for the first time, the impacts of two categories of MHWs, one with moderate and another with extreme intensity, on the physiology and PST production of *G. catenatum*.

## 2. Results

### 2.1. Cell Count, Size, and Photosynthetic Efficiency

Exposure to different temperature treatments did not elicit significant changes in the cell concentration of *G. catenatum* cultures (Figure 1a, Table 1 and Appendix A). However, the number of cells in each culture increased significantly throughout the exposure to all treatments (Figure 1b, Table 1 and Appendix A).

From Figure 2a (Table 1 and Appendix A), it can be observed that the category IV MHW elicited significant effects on the probability of forming chains, promoting the occurrence of single cells in *G. catenatum* cultures. On the other hand, all three stages of MHWs elicited significant changes in this trait (Figure 2b), the most notable being an increase in the formation of chains between the beginning and latter stages of both MHWs (Table 1 and Appendix A).

The different temperature treatments did not significantly affect the length of the individual cells (Figure 3a, Table 1 and Appendix A). However, exposure stages impacted the length, causing a decrease in cell size at the peak of both MHWs (Figure 3b, Table 1 and Appendix A).

Regarding the photosynthetic efficiency, from Figure 4a we can observe that temperature treatments did not affect the maximum quantum yield of photosystem II (PSII, Table 1 and Appendix A). Conversely, there was an increase in the Fv/Fm ratio throughout the course of the simulated MHWs (Figure 4b, Table 1 and Appendix A). Correlation plots between cell parameters can be found in Appendix A.

### 2.2. Toxin Concentration and Profile

The PST toxicity was negatively affected by the temperature treatment to which *G. catenatum* cultures were exposed (Figure 5a, Table 1 and Appendix A), and toxin content significantly varied between the different phases of the MHW with a decrease of PSP concentration from the beginning towards the recovery stage (Figure 5b, Table 1 and Appendix A).

Regarding the toxin profile present in the cultures, C4 was the most abundant toxin found (Figure 6), followed by C1 and C2. Notably, dcGTX3, dcSTX, C1, C2, and C4 content (in fM) decreased in category I MHW when compared to the control. In some cases (dcSTX and C4), there was a similar decrease in the compound’s content between the control and the most severe MHW (category IV). When analysing the MHWs in the different stages (Figure 7), a similar trend can be found, with C4, C1, and C2 being the most abundant toxins. There is also a general decrease in any given analogue between the beginning and later stages of the simulated MHWs (Table 1 and Appendix A). Correlation plots between toxin content and cell parameters can be found in Appendix A.

## 3. Discussion

The present study constitutes the first attempt at understanding how MHWs can affect the growth, photosynthetic efficiency, and toxin production of *G. catenatum,* a PST-producing dinoflagellate with a temperate to tropical distribution. The results presented herein suggest that *G. catenatum* populations from temperate areas (Portugal, NE Atlantic) may be resilient to abrupt and severe changes in water temperature such as those induced by MHWs. After 30 days of exposure to fluctuating temperatures reaching up to +1.4 (MHW cat I) and 5.6 °C (MHW cat IV) compared to control temperatures, most parameters tested were not significantly affected. Usually, higher temperatures tend to increase metabolic rate [21], promoting cell division and leading to higher cell concentrations. However, there is growing evidence that such responses are very dependent on latitude and nutrient availability [7,21]. Previous literature on *G. catenatum* exposed to different temperatures in laboratory conditions indicated that the optimal temperature for this species was between 20 and 30 °C [15,16], depending on the site of origin of the strains. *Gymnodinium catenatum* is a chain-forming species, and, under laboratory conditions, chain formation and length are considered indicators of optimal growth conditions [16] and references therein. The findings presented by Band-Schmidt et al. [16], where cells had their optimal growth rates in the laboratory around 24 °C (with environmental temperatures ranging from 20 to 21.9 °C), showed that chain length also reached its maximum at this temperature. In the present study, there was a significant decrease in the ability of the cells to form chains upon exposure to the most severe MHW (24.1 °C), suggesting that cells were under suboptimal conditions and stress despite other physiological parameters suggesting otherwise. In this study, we argue that the shortening of the chain length may be a more sensitive proxy for the effect of MHWs in *G. catenatum* populations. The ecological significance of this effect on *G. catenatum* populations may be severe since chain formation is associated with an increase in swimming capacity, an important trait for niche exploration in more turbulent waters such as the upwelling areas where this species occurs [22].

Atkinson et al. [23] showed that with an increase in 1 °C body size decreases by 2.5% in dinoflagellates and other protists; this was not observed in the present study, where an increase of 1.4 and 5.6 °C did not elicit significant changes in cell size. These authors argued that this rule can be verified when there is resource limitation or reduction, such as decreased gas diffusion rates with increased temperature, and that upon removal of resource limitation, there can be exceptions. In the present study, our cultures were daily replenished with fresh growth medium to maintain seawater conditions as stable as possible, which probably relieved some nutrient limitations.

Photosynthesis is highly dependent on temperature, with photosynthetic efficiency increasing at higher temperatures until an optimal threshold is met, from which further increases in temperature result in decreased photosynthetic rates [24]. The present study, yet again, points to the temperature treatments having no detectable effects on the light-harvesting systems of *G. catenatum* populations. Nevertheless, it should be denoted that this does not necessarily imply the existence of a temperature-insensitive mechanism in this species/strain, as the light-harvesting mechanism accounts only for half of the whole photosynthetic process, leaving aside the carbon concentration metabolism. Upon exposure to an environmental stressor—e.g., nutrient limitation—phytoplankton species appear to change the number and size of the photosystems involved in photosynthesis, a typical compensatory feedback mechanism that allows cells to maintain the PSII quantum efficiency [21,25]. Additionally, Fernández-González et al. [7] showed that Fv/Fm ratios along different latitudes (~30° N to ~30° S) do not present clear trends and are independent of nutrient limitation and temperature, suggesting that similar processes could possibly be occurring in the cultures used in the present study.

Temperature effects on toxin production are very variable, depending on the species and even within the species—depending, for instance, on the latitude of each strain/population [26]. The information in the literature regarding the effects of temperature increase on the PST production of *G. catenatum* is contradictory. While some conclude that higher temperatures stimulate toxin production [16], others verify the opposite: that toxin production is maximal at lower temperatures [27,28]. In this study, we observed that with increasing temperatures PST production decreased significantly between control conditions and MHW treatments and there were qualitative changes in the toxin profile between treatments and MHW stages. This species can produce up to 21 saxitoxin (STX) derivatives [15], depending on several factors. In fact, the toxin profile varies greatly among and between populations due to genetic differences, leading to different expressions in the enzymes involved in toxin synthesis [19]. Here we found that the studied isolate produced STX derivatives similar to those found in the literature for other strains from the same region [15,29,30], and that, in general, MHW treatments decreased the production of STX derivatives when compared to control conditions, especially throughout the exposure.

It is known that extreme events, such as MHWs can exacerbate the impacts of other oceanic climate-related variables (like ocean warming) on marine biota [5]. MHWs have been shown to trigger massive HAB events that have severe consequences [10,11]. This study constitutes the first laboratory-simulated MHW using a PST producer with a wide distribution in temperate and tropical habitats. Our results indicate that *G. catenatum* has a high tolerance for MHWs, which is accompanied by lowered toxin production at higher temperatures. The increase in frequency, intensity, and duration of MHWs may lead to reduced severity of *G. catenatum* toxic blooms, in contrast to what has been reported for other HAB species. Nevertheless, this study focused exclusively on a single stressor (temperature), and there’s a body of information [18,31,32,33] showing that algal growth and toxin production are also dependent on nutrient availability and microbial associations, among other external dynamic factors (e.g., pH, stratification, UV radiation). In marine systems worldwide, different combinations of these factors are occurring simultaneously, shaping community interactions according to their species composition and ocean dynamics. Therefore, the inclusion of more variables in studies focusing on HAB growth and toxicity is of paramount importance. More studies are needed to better understand how we can expect MHWs to impact this species, especially including prolonged MHWs and additional factors, as no single factor acts independently in the environment.

## 4. Materials and Methods

### 4.1. Strain Origin and Laboratory Acclimation

The clonal strain of *G. catenatum* (IO13-04) used in the present study was established through the isolation of a single cell from phytoplankton net samples collected in Espinho (41°0′0″ N 8°38′56.4″ W), on the western coast of Portugal, in November 2005. Once established, cultures were maintained in L1 medium [33] salinity 33 ppt, at 19.0 ± 1.0 °C, and under 12 h:12 h light: dark cycle and photosynthetic photon flux density (PPFD) of ca. 40 μmol photons m^−2^s^−1^. Cultures were maintained in the algae culture collection at Lisbon University (ALISU).

To obtain the necessary inoculum for the initiation of the planned experiments, cultures were scaled up at the facilities of Laboratório Marítimo da Guia (Cascais, Portugal). In this study, cultures were acclimated to the control experimental conditions for two months in temperature-controlled (STC-3000 temperature controllers, hysteresis 0.3 °C) water baths at 18.5 °C (control temperature estimated climatological mean for the Western coast of Portugal) with 10:14 L:D photoperiod under AquaRay GroBeam lamps (600 ultima, TMC, Manchester, UK). A culture medium (L1) was added weekly and cell counts were carried out frequently to ensure that the cultures were kept at an exponential growth phase.

### 4.2. Experimental Design

Following acclimation, all the cultures were combined and re-distributed between experimental treatments in 2 L Schott flasks as follows: (i) control temperature (18.5 °C, n = 8 flasks), (ii) MHW category I (19.9 °C, n = 8), and (iii) MHW category IV (24.1 °C, n = 8). To simulate MHWs, a 30-year (1989–2019) dataset for seawater surface temperature from the western coast of Portugal was obtained from NOAA [accessed on 15 January 2021]. With this dataset, it was determined that the average temperature throughout the 30 years was 18.5 °C. In R, the intensity and rates of onset and offset of MHWs were determined through the package ‘heatwaveR’ [34]—which uses the definition of MHWs according to [4]. Table 2 summarises the details pertaining to the simulation of MHWs. In this study, category I MHW was determined to have a peak intensity of 19.9 °C when climatology was 18.5 °C, whereas category IV MHW was determined to have a peak intensity of 24.1 °C, with the same climatology.

The duration of 30 days corresponds to the following phases: (i) 10 days of onset temperature increasing slope, (ii) 10 days in peak temperature, and (iii) 10 days in offset temperature decreasing slope. The duration was chosen to accompany the cultures’ growth curve (circa 10 days) and the definition of MHW: “MHW needs to persist for at least five days” [3].

### 4.3. Culture Maintenance and Sample Acquisition

During exposure to MHWs, daily additions of small amounts (50 mL) of fresh L1 growth medium [35] were carried out to avoid sudden changes in water quality. Since the cultures were immersed in the circulating water bath, temperature parameters were obtained from the water bath and not directly from the cultures to avoid contamination. Every three days, samples were collected from four random replicates in each treatment for cell counts, cell size, and photobiological measurements. Every ten days, half of the culture volume (500 mL) was collected for toxin analysis, corresponding to the amount of medium added in the 10-day period. Before sampling, cultures were always carefully homogenised by gently swirling the flasks. Cell counts were carried out in a Sedgewick-Rafter counting chamber, where the total number of single cells (1 cell) and the number of cells in chains (2 cells or more) were registered. Photographs (n = 100 per treatment) were taken with a stereomicroscope (Leica S APO coupled with MC 190 HD camera, Leica Microsystems) to carry out cell size (i.e., cell height) measurements in ImageJ. Photobiological parameters (basal fluorescence (Fo), maximum fluorescence (Fm), and maximum quantum PSII yield (Fv/Fm)) were obtained using a Water-PAM chlorophyll fluorometer (Heinz Walz GmbH, Pfullingen, Germany) in dark-acclimated (15 min.) cultures. The volume extracted from each replicate was filtered onto 47 mm Whatman GF/C with a nominal pore size of 1.2 µm under a low vacuum and stored at −80 °C for toxin quantification.

### 4.4. Toxin Extraction and Quantification

#### 4.4.1. Extraction

*Gymnodinium catenatum* toxin extraction was performed following [30] with slight modifications. Toxins were extracted in 4 mL of 0.05 M acetic acid and sonicated for 1 min at 25 W and 50% pulse duty cycle (Vibracell, Sonic & Materials, Newtown, CT, USA) in an ice bath. The extract was then centrifuged (3000× *g*) for 10 min, and the supernatant was collected.

#### 4.4.2. Solid-Phase Extraction (SPE) Clean-Up

The supernatant was cleaned by solid-phase extraction (SPE) following [36], with slight modifications. Briefly, 1 mL of the acetic acid extract was transferred to a polypropylene centrifuge tube and 5 µL of NH_4_OH was added.

The SPE procedure was performed on an SPE Vacuum Manifold with amorphous graphitised polymer carbon Supelco ENVI-Carb 250 mg/3 mL cartridges (P/N:57088, Sigma–Aldrich, Algés, Portugal). The ENVI-Carb cartridges were conditioned with 3 mL of acetonitrile/water/acetic acid (20:80:1 *v*/*v*/*v*), followed by 3 mL of water/NH_4_OH (1000:1 *v*/*v*), with both solutions eluting to waste. From the acetic acid extracts/NH_4_OH solution, 400 µL were loaded onto the conditioned cartridges and were washed with 700 µL of water, both eluting to waste.

The toxins were then eluted with 2 mL of acetonitrile/water/acetic acid (20:80:1 *v*/*v*/*v*), into a polypropylene test tube. The eluate was transferred to a polypropylene autosampler vial and diluted with acetonitrile before analysis.

#### 4.4.3. LC-HRMS Analysis

The samples were analysed by liquid chromatography-high resolution mass spectrometry (LC-HRMS). Chromatographic separation was carried out using an UltiMate 3000 UHPLC system coupled to an Orbitrap Elite mass spectrometer (Thermo Scientific, Bremen, Germany) equipped with a heated electrospray ionization source (HESI-II). The PST analogues were separated using an ACQUITY Premier BEH Amide (2.1 × 100 mm, 1.7 μm, Waters, Milford, MA, USA) at 35 °C. Samples were held in the autosampler at 4 °C. The mobile phase was composed of water with 0.1% formic acid and 10 mM ammonium formate (A) and acetonitrile with 0.1% formic acid and 2% 10 mM ammonium formate solution (B). The gradient (in *v/v* %) started with 5% of B and increased linearly to 95% in 11 min. This composition was maintained for 1 min and then returned to 5% of B in 1 min and maintained at this composition for 2 min before the next run [37]. The flow rate was 0.3 mL min^−1^ and the injection volume was 10 µL. Data were acquired under positive (ESI+) and negative (ESI-) polarity using the following ionization parameters: spray voltage, 3.8 kV; sheath gas, 40 arbitrary units; auxiliary gas, 10 arbitrary units; heater temperature, 300 °C; capillary temperature, 325 °C; and S-Lenses RF level, 69.10%. The LC-HRMS acquisition was performed under multiple-reaction monitoring (MRM) mode. A minimum of two MRM transitions for each PST analogue were selected based on MS/MS spectra collision-induced dissociation (CID), acquired by infusing the individual standards into the mass spectrometer. The collision energy used was 35 arbitrary units. Positive mode (ESI+) transitions were used exclusively for STX, NEO, dcSTX, and dcNEO quantitation. Negative mode (ESI-) transitions were used for the remaining analogues (GTX1to 6, dcGTX2 and 3, and C1 to 4) quantitation. The MRM transitions used are summarized in Appendix A. Quantitation was performed by running calibration curves, with five levels of different concentrations of PST analogues standards mixture, in triplicate. The correlation coefficient (R^2^) was determined using a linear regression model. The toxicity equivalency factors (TEF) used were taken from those recommended by European Food Safety Authority—EFSA [38]. The limits of detection (LOD) and quantification (LOQ) were calculated from the standard deviations (SDs) obtained after five injections of the second-lowest concentration of the standard curve (3 × SD and 10 × SD, respectively, Appendix A). Certified reference materials (CRM): N-sulfocarbamoyl gonyautoxin-2 (C1 40.1 ± 2.4 μg/g), N-sulfocarbamoyl gonyautoxin-3 (C2 11.5 ± 0.9 μg/g), N-sulfocarbamoyl gonyautoxin-1 (C3 12.6 ± 0.9 μg/g), N-sulfocarbamoyl gonyautoxin-4 (C4 3.4 ± 0.3 μg/g), Gonyautoxin-1 (GTX1 27.3 ± 1.6 μg/g), Gonyautoxin-2 (GTX2 22.2 ± 1.5 μg/g), Gonyautoxin-3 (GTX3 8.2 ± 0.6 μg/g), Gonyautoxin-4 (GTX4 7.3 ± 0.6 μg/g), Gonyautoxin-5 (GTX5 18.1 ± 1.2 μg/g), Gonyautoxin-6 (GTX6 10.0 ± 0.5 μg/g), Decarbamoylgonyautoxin-2 (dcGTX2 35.1 ± 1.9 μg/g), Decarbamoylgonyautoxin-3 (dcGTX3 8.0 ± 0.9 μg/g), Neosaxitoxin dihydrochloride (NEO 20.3 ± 1.2 μg/g), Saxitoxin dihydrochloride (STX 20.3 ± 1.3 μg/g), Decarbamoylneosaxitoxin dihydrochloride (dcNEO 9.1 ± 0.5 μg/g), and Decarbamoylsaxitoxin dihydrochloride (dcSTX 19.5 ± 1.7 μg/g) were purchased from CIFGA Laboratories S.A. (Lugo, Spain).

### 4.5. Data Analysis

To confirm the influence of the temperature treatments on cell parameters and toxin production, generalized linear models (GLM, “glm” function) were fitted to the response data [39]. Treatments were set as a factor with three levels (i.e., Control, MHW I, and MHW IV), replicates as a factor with four levels, and MHW stage as a factor with three levels. A GLM from the Poisson family (log link) was fitted to cell counts, and the binomial family (i.e., using the logit link) was used to fit chain formation data (0/1). Lastly, linear models (i.e., using the identity link) were fitted to cell length, photosynthesis, and toxin concentrations. Type II Wald chi-square tests (function “Anova”) were performed on the models to evaluate the effect of the different treatments on the response variables. The model assumptions of normality, homoscedasticity of residuals, as well as independence between data points and influential observations were checked. Post-hoc comparisons (function “emmeans” in the “emmeans” package) were performed on the temperature treatment and MHW stage whenever an influence of these variables was detected by the Wald chi-squared test. Significance levels were set at *p* < 0.05, and *p*-values were adjusted through Tukey corrections in order to avoid type I errors. Statistical analyses were carried using the statistical and programming software R 4.2.2 [40] (R Core Team, 2022).

## Figures and Tables

**Figure 1 toxins-15-00157-f001:**
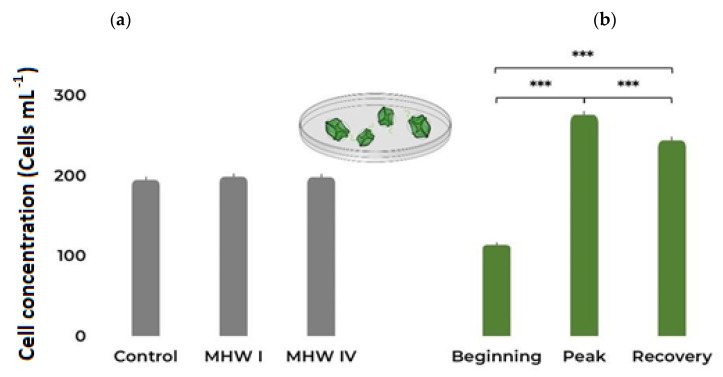
*Gymnodinium catenatum* cell concentrations under (**a**) control and MHW categories I and IV treatments, taking into account all stages; (**b**) the different MHW stages, i.e., beginning, peak, and recovery, taking into account all treatments. Bars represent the estimated marginal means (EMM) ± SE, under the Poisson GLM (averaged over the heatwave stage and heatwave category, respectively). Comparison significance levels: *** *p* < 0.001.

**Figure 2 toxins-15-00157-f002:**
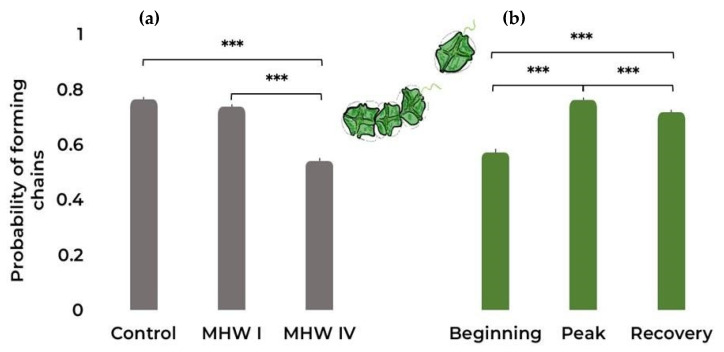
*Gymnodinium catenatum* chain formation under (**a**) control and MHW categories I and IV treatments, taking into account all stages; (**b**) the different MHW stages: i.e., beginning, peak, and recovery, taking into account all treatments. Bars represent the probability ± SE as predicted by the binomial GLM (averaged over the replicate and heatwave stage and heatwave category, respectively). Comparison significance levels: *** *p* < 0.001.

**Figure 3 toxins-15-00157-f003:**
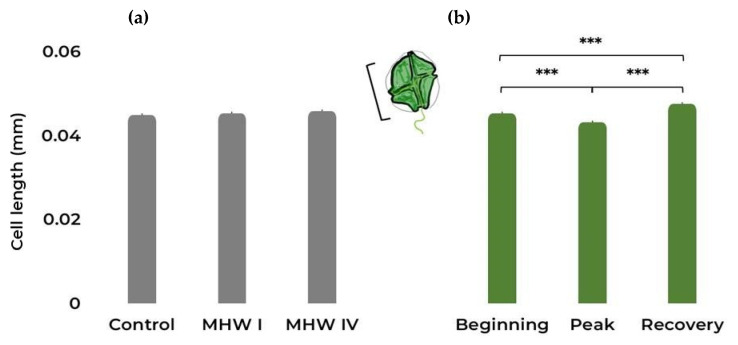
*Gymnodinium catenatum* cell length (mm). (**a**) Control and MHW categories I and IV treatments, taking into account all stages; (**b**) the different MHW stages: i.e., beginning, peak, and recovery, taking into account all treatments. Bars represent the estimated marginal means (EMM) ± SE, under the gaussian GLM (averaged over the replicate and heatwave stage and heatwave category, respectively). Comparison significance levels: *** *p* < 0.001.

**Figure 4 toxins-15-00157-f004:**
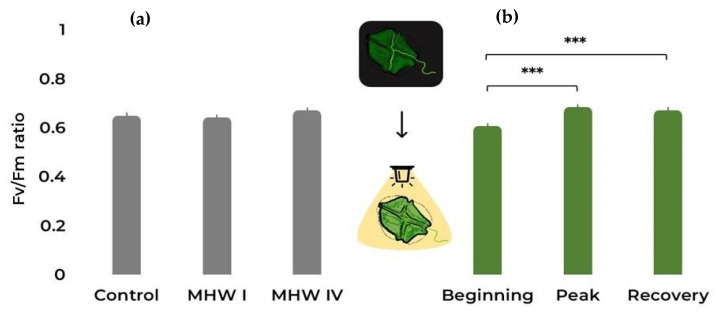
*Gymnodinium catenatum* Fv/Fm under (**a**) control and MHW categories I and IV treatments, taking into account all stages; (**b**) the different MHW stages: i.e., beginning, peak, and recovery, taking into account all treatments. Bars represent the estimated marginal means (EMM) ± SE, under the gaussian GLM (averaged over the replicate and heatwave stage and heatwave category, respectively). Comparison significance levels: *** *p* < 0.001.

**Figure 5 toxins-15-00157-f005:**
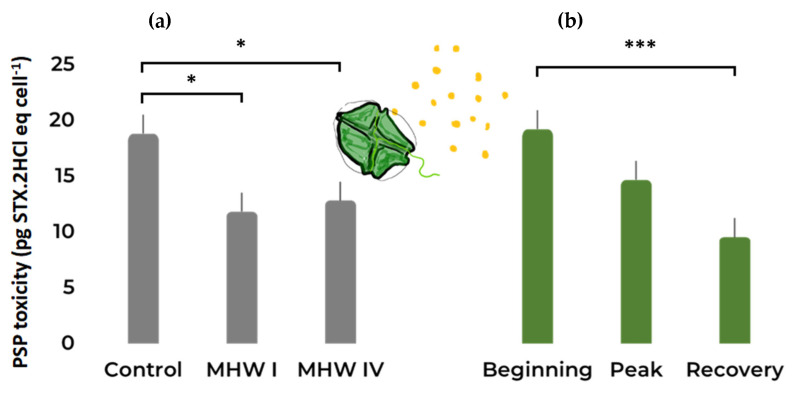
*Gymnodinium catenatum* PSP toxicity under (**a**) control and MHW categories I and IV treatments, taking into account all stages; (**b**) the different MHW stages: i.e., beginning, peak, and recovery, taking into account all treatments. Bars represent the estimated marginal means (EMM) ± SE, under the gaussian GLM (averaged over the replicate and heatwave stage and heatwave category, respectively). Comparison significance levels: * *p* < 0.05, *** *p* < 0.001.

**Figure 6 toxins-15-00157-f006:**
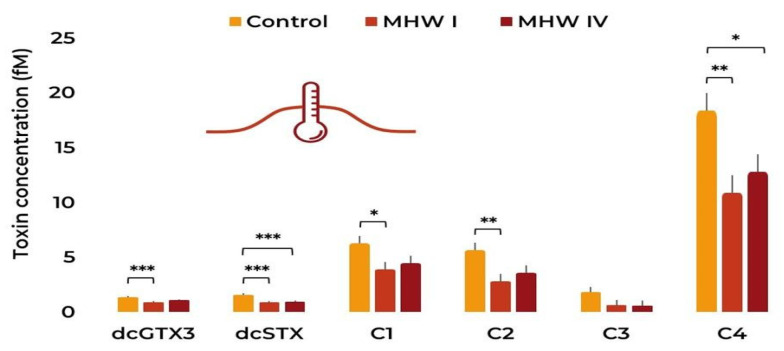
*Gymnodinium catenatum* PST profile under the control and MHW categories I and IV treatments. Bars represent the estimated marginal means (EMM) ± SE, under the gaussian GLM (averaged over the replicate and heatwave stage). Comparison significance levels: * *p* < 0.05, ** *p* < 0.01, *** *p* < 0.001.

**Figure 7 toxins-15-00157-f007:**
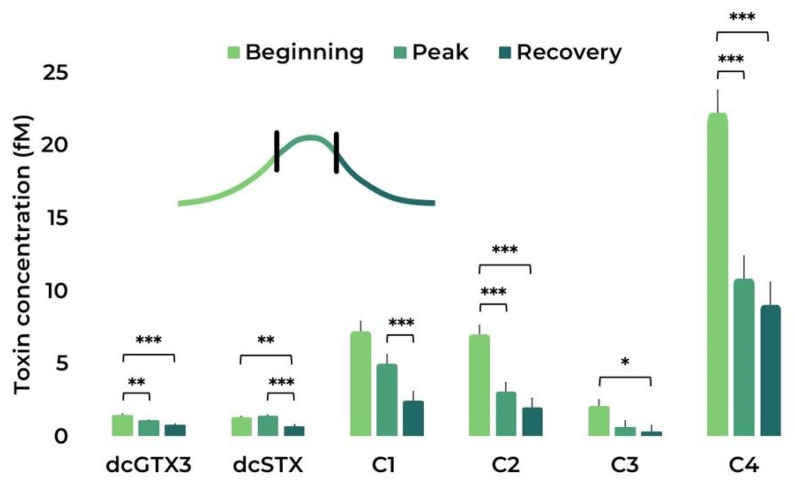
Paralytic shellfish toxins’ profile during the MHW stages: beginning, peak, and recovery. Bars represent the estimated marginal means (EMM) ± SE, under the gaussian GLM (averaged over the replicate and treatment). Comparison significance levels: * *p* < 0.05, ** *p* < 0.01, *** *p* < 0.001.

**Table 1 toxins-15-00157-t001:** Results from generalized linear models (Wald Chi-squared Tests, function Anova), depicting the effect of predictor variables (Treatment, sampling date, replicate, and MHW stage) on *Gymnodinium catenatum* chain formation, cell length, photophysiology (Fv/Fm), PSP toxicity, and toxin concentration (dcGTX3, dcSTX, C1, C2, C3, C4). C—control, I—marine heatwave category I, IV—marine heatwave category IV, fM—concentration in fM per cell, MF—molar fraction (%).

Response	Model Distribution	Predictor	χ^2^	Degrees of Freedom	*p*-Value
Cell concentration	Poisson	Treatment	6.08	2	**0.048**
		Stage	917.9	2	**<0.001**
		Treatment:Stage	65.2	4	**<0.001**
Chain formation	Binomial	Treatment	390.0	2	**<0.001**
		Stage	145.8	2	**<0.001**
		Replicate	70.1	3	**<0.001**
		Treatment:Date	56.2	4	**<0.001**
Cell length	Gaussian	Treatment	2.75	2	0.253
		Stage	65.16	2	**<0.001**
		Treatment:Stage	86.07	4	**<0.001**
Fv/Fm	Gaussian	Treatment	3.949	2	0.139
		Stage	29.754	2	**<0.001**
PSP toxicity	Gaussian	Treatment	9.814	2	**0.007**
		Stage	16.21	2	**<0.001**
dcGTX3 (fM)	Gaussian	Treatment	13.33	2	**0.001**
		Stage	29.44	2	**<0.001**
dcGTX3 (MF)	Gaussian	Treatment	0.66	2	0.718
		Stage	27.29	2	**<0.001**
		Treatment:Stage	9.18	4	0.057
dcSTX (fM)	Gaussian	Treatment	19.49	2	**<0.001**
		Stage	17.84	2	**<0.001**
dcSTX (MF)	Gaussian	Treatment	6.55	2	**0.038**
		Stage	9.51	2	**0.009**
C1 (fM)	Gaussian	Treatment	6.39	2	**0.041**
		Stage	23.97	2	**<0.001**
C1 (MF)	Gaussian	Treatment	2.36	2	0.308
		Stage	12.99	2	**0.002**
		Treatment:Stage	9.80	4	**0.044**
C2 (fM)	Gaussian	Treatment	9.44	2	**0.008**
		Stage	30.37	2	**<0.001**
C2 (MF)	Gaussian	Treatment	6.55	2	**0.038**
		Stage	9.51	2	**0.009**
C3 (fM)	Gaussian	Treatment	4.38	2	0.112
		Stage	7.99	2	**0.018**
C3 (MF)	Gaussian	Treatment	4.65	2	0.098
		Stage	4.08	2	0.130
C4 (fM)	Gaussian	Treatment	12.16	2	**0.002**
		Stage	40.50	2	**<0.001**
		Treatment:Stage	9.44	4	0.051
C4 (MF)	Gaussian	Treatment	1.02	2	0.602
		Stage	18.06	2	**<0.001**

*p*-values < 0.05 are represented in bold.

**Table 2 toxins-15-00157-t002:** Exposure of *Gymnodinium catenatum* to simulated MHW categories I and IV.

	Category I	Category IV
Peak intensity (°C)	19.9 °C	24.1 °C
Rate of onset (°C day^−1^)	0.13	0.55
Duration (days)	10	10
Rate of offset (°C day^−1^)	0.14	0.56
Total duration (days)	30	30

## Data Availability

The data has been made available in the Appendix A.

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
