# Peer review of "Gymnodinium catenatum Paralytic Shellfish Toxin Production and Photobiological Responses under Marine Heat Waves"

_toxins, 2023, doi:10.3390/toxins15020157_

Round 1

Reviewer 1 Report

General comments:

The manuscript focuses on the response of PST production and photobiology in the dinoflagellate Gymnodinium catenatum under marine heat waves. The main conclusion of this manuscript is the increase in frequency, intensity, and duration of MHWs may lead to reduced severity of G. catenatum toxic blooms. It is an interesting conclusion since many published studies show negative effects (increased cell density, recurrence and toxicity) of climate change, such as atmospheric-oceanographic anomalies.

I have some comments that can help improve the manuscript (see specific commentaries). This paper provides valuable information that helps explain the potential response of an important toxic microalgae, G. catenatum, to different temperature scenarios under Marine Heat Waves in the context of global climate change. Therefore, I recommend a minor revision.

Specific commentaries:

Introduction

Page 1 (lines 34-35). Review the format for citing references. “Hobday and colleagues [4]” Should say “Hobday et al. [4]”.

Page 2 (lines 49). …on the human economy and populations [++]. Include a reference.

Material and methods

Page 8 (line 265). …definition of MHW: “(...) MHW needs to…”. Is something missing in the parenthesis?

Page 9 (lines 348-349). RStudio is just a visualizer for R. The correct reference for R software is:

39. R Development Core Team. R: A language and environment for statistical computing. R Foundation for Statistical Computing, Vienna, Austria. ISBN 3-900051-07-0, URL http://www.r-project.org/. 2022.

Statistical analyses were carried out using were performed using the statistical and programming software R 4.2.2 [39]

Results

Page 2 (line 82). Table S1 is a very important result of their work to be part of the supplementary material. Move the table to the results section.

Page 3 (line 101). Replace “affect” with “effect”.

Page 4 (line 124). G. catenetum in italic.

Page 5 (line 138). Delete “t” after control.

Discussion

The discussion is very well written. However, considering the relevance of the results, it would be appropriate to include a small section of conclusions.

Author Response

Reviewer #1

General comments:

The manuscript focuses on the response of PST production and photobiology in the dinoflagellate Gymnodinium catenatum under marine heat waves. The main conclusion of this manuscript is the increase in frequency, intensity, and duration of MHWs may lead to reduced severity of G. catenatum toxic blooms. It is an interesting conclusion since many published studies show negative effects (increased cell density, recurrence and toxicity) of climate change, such as atmospheric-oceanographic anomalies.

I have some comments that can help improve the manuscript (see specific commentaries). This paper provides valuable information that helps explain the potential response of an important toxic microalgae, G. catenatum, to different temperature scenarios under Marine Heat Waves in the context of global climate change. Therefore, I recommend a minor revision.

Reply: The authors would like to thank the reviewer for his/her comments and suggestions, and for the time taken for this revision.

Specific commentaries:

Introduction

  • Page 1 (lines 34-35). Review the format for citing references. “Hobday and colleagues [4]” Should say “Hobday et al. [4]”.

Reply: Thank you for your suggestion, the text was changed accordingly (L. 35).

  • Page 2 (lines 49). …on the human economy and populations [++]. Include a reference.

Reply: The latest report from WHOI was added as a reference, reference [9]: (https://www.whoi.edu/wp-content/uploads/2022/01/2022WHOI-HarmfulAlgalBlooms-Report.pdf) (L. 48).

Material and methods

  • Page 8 (line 265). …definition of MHW: “(...) MHW needs to…”. Is something missing in the parenthesis?

Reply: The parenthesis served to indicate that the sentence had more information before what was extracted. To avoid confusion, the parenthesis were removed.

  • Page 9 (lines 348-349). RStudio is just a visualizer for R. The correct reference for R software is:
  1. R Development Core Team. R: A language and environment for statistical computing. R Foundation for Statistical Computing, Vienna, Austria. ISBN 3-900051-07-0, URL http://www.r-project.org/. 2022.

Statistical analyses were carried out using were performed using the statistical and programming software R 4.2.2 [39]

Reply: Thank you for your input, we have changed the text and references accordingly. This reference is now reference [40] and can be found on line 377 and listed on line 493-494.

Results

  • Page 2 (line 82). Table S1 is a very important result of their work to be part of the supplementary material. Move the table to the results section.

Reply: Table S1 was moved to the result section, being now Table 1, found on line 92. The order of the remaining supplemental tables was arranged accordingly.

  • Page 3 (line 101). Replace “affect” with “effect”.

Reply: We would like to politely argue that, in this case, affect is the most appropriate word, as a verb. We mean to say that the temperature did not elicit a response in cell length.

  • Page 4 (line 124). G. catenetum in italic.

Reply: Thank you for noticing, the error was corrected.

  • Page 5 (line 138). Delete “t” after control.

Reply: Thank you for noticing, the error was removed (L. 166).

Discussion

The discussion is very well written. However, considering the relevance of the results, it would be appropriate to include a small section of conclusions.

Reply: We thank the reviewer for this comment. While we agree that a Conclusions section would be better as a way of structuring the manuscript, the Toxins journal format does not allow for such a section to be included. Notwithstanding, the last paragraph of the Discussion acts as a global conclusion for the manuscript (see L. 228-245). 

Reviewer 2 Report

Dear Editor, I have read in detail the article entitled: "Gymnodinium catenatum paralytic shellfish toxin production and photobiological responses under marine heat waves".

The following are my comments on the article:

Abstract:

Please be more precise about the topics that should be detailed in this section. Please review the details of the journal.

Introduction:

Be more precise about the objective of the study and state the hypothesis of the study.

Line 65: this sentence should be better argued from the point of view of the references and also in the text.

Results:

Please be more specific in the results Figure 1B. What is the purpose of this figure?

Figure 1A Why were cells exposed only to MHW I-IV?

Figure 5A what is the toxicity of G. catenatum. it would seem that micrograms per kilogram is too much.

Figure 6. was only dcGTX3 detected?

Why was dcGTX2 not detected? These analogs are epimers and the yield between the analogs must maintain a ratio. It is likely that your method is under the LOD of dcGTX2.

Materials and methods

What other chemical parameters were analyzed during the MHW exposure?

What was the density of microalgae used in the experiments?

4.4.3. LC-HRMS analysis

Please include the LOD and LOQ of the toxins.

Also include where you obtained the certified standards for the analyses.

Discusion.

It is relatively well addressed with respect to the results obtained, but please be more precise in the argumentation of the parameters that you have not considered in this study, since it is very difficult to understand the proposal when considering that you have left out parameters that can produce a Bloom.

Author Response

Reviewer #2

Dear Editor, I have read in detail the article entitled: "Gymnodinium catenatum paralytic shellfish toxin production and photobiological responses under marine heat waves".

The following are my comments on the article:

Abstract:

  • Please be more precise about the topics that should be detailed in this section. Please review the details of the journal.

Reply: We thank the reviewer for their time in revising the present manuscript. Regarding the present comment, the abstract has been divided into specific sections: background (L. 5-10); methodology (L. 10-12); results (L. 12-14) and finally discussion and conclusions (L. 14-18).

Introduction:

  • Be more precise about the objective of the study and state the hypothesis of the study.

Reply: We changed the last paragraph of the introduction to read as follows (L. 71-77). We hope it is now clearer.

“Understanding the potential impacts of short-term extreme events, such as MHWs is of vital importance, given the reduced opportunity for species adaptation and the particularly severe repercussions that these events can have on marine ecosystems [3,20] and human populations. The hypothesis postulated here is that MHWs will affect G. catenatum growth and toxin production. Therefore, the aim of the present study was to investigate, for the first time, the impacts of two categories of MHWs, one with moderate and other with extreme intensity, on the physiology and PST production of G. catenatum.”

  • Line 65: this sentence should be better argued from the point of view of the references and also in the text.

Reply: The section was re-written, and can be found on lines 64 through 70 as follows:

“Throughout the years, several studies have investigated the effects of temperature on the growth rates, life cycle transitions and toxin profiles of G. catenatum [16,18,19]. They were carried out using isolates from different areas, with the objective of using toxin profiles as biochemical markers to distinguish populations and provide knowledge on the thermal range this species can inhabit.  To date, no study has investigated the impact of predicted temperature increases related to climate change or MHW exposure.”

Results:

  • Please be more specific in the results Figure 1B. What is the purpose of this figure?

Reply: Figure 1B represents the cell density on each stage of the MHW taking into account all the treatments. To prevent showing a very busy figure (which would include all three treatments and three MHW stages), we opted to combine all stages and treatments. In short, this means that the first bar of Figure 1B represents all the data at the beginning of the MHWs combined, including data for control, MHW category I and MHW category IV. This was done so that the potential effects of stage exposure were visible.

  • Figure 1A Why were cells exposed only to MHW I-IV?

Reply: Due to spatial/logistical constraints, and to prevent a very large number of replicates, some decisions had to be taken, namely reducing the number of temperature treatments. Therefore, we chose the mildest and the most severe MHW categories, since the mildest has already been registered in the past for several areas of the Portuguese coast, and the most severe category is a proxy to a likely worst-case future scenario.

  • Figure 5A what is the toxicity of G. catenatum. it would seem that micrograms per kilogram is too much.

Reply: We thank the reviewer for this comment, as it lead us to double check, finding that, indeed the calculations were not complete. Figure 5 now is expressed as pg STX. 2HCL eq cell-1, as measurements were made with cell concentrations rather than weight. This new addition changed the figure and the signficance of this section of the results.

  • Figure 6. was only dcGTX3 detected? Why was dcGTX2 not detected? These analogs are epimers and the yield between the analogs must maintain a ratio. It is likely that your method is under the LOD of dcGTX2.

Reply: Indeed this was the case, the information about LOD and LOQ was added to the supplementary material (Table S3). 

Materials and methods

  • What other chemical parameters were analyzed during the MHW exposure?

Reply: No chemical parameters other than the ones mentioned in the present manuscript were analyzed.

  • What was the density of microalgae used in the experiments?

Reply: All densities are presented in Figure 1 A and B as cell concentrations (cells ml-1).

  • 4.3. LC-HRMS analysis. Please include the LOD and LOQ of the toxins. Also include where you obtained the certified standards for the analyses.

Reply: LOD and LOQs were added to the supplementary material and are now part of Table S3. A sentence on how these were calculated was included in L. 348-350. The information regarding the certified standards was also added to the text, on L. 350-360.

Discusion.

It is relatively well addressed with respect to the results obtained, but please be more precise in the argumentation of the parameters that you have not considered in this study, since it is very difficult to understand the proposal when considering that you have left out parameters that can produce a Bloom.

Reply: We agree with reviewer #2, that there are many factors influencing bloom formation and toxin production. We mention some of the aspects that were not addressed in the present paper that would have been interesting to address in future occasions. In fact, the actual trigger(s) of harmful algal blooms are region and species-specific, and among them are species composition, water body dynamics, etc. All of these variables and more are acting simultaneously to produce the ideal conditions for a certain HAB species to proliferate. We made the text clearer on this matter, and you can find the following rewritten sentences on lines 236 through 243:

“Nevertheless, this study focused exclusively on a single stressor (temperature), and there’s a body of information [18,31–33] showing that algal growth and toxin production are also dependent on nutrient availability and microbial associations, among other external dynamic factors (e.g. pH, stratification, UV radiation). In marine systems worldwide, different combinations of these factors are occurring simultaneously, shaping community interactions according to their species composition and ocean dynamics. Therefore, it is of paramount importance the inclusion of more variables in studies focusing on HAB growth and toxicity.”

Reviewer 3 Report

The manuscript entitled “Gymnodinium catenatum paralytic shellfish toxin production and photobiological responses under marine heat waves” aimed to investigate how marine heatwaves impact parameters relating to G. catenatum cell growth and toxin production using multi-biomarker approach. Authors concluded that production of paralytic shellfish toxins by G. catenatum does not seem to be potentiated under marine heatwave exposure.

There is a very relevant new document regarding the threat of harmful algal blooms that can be downloaded on the Woods Hole Oceanographic Institution: https://www.whoi.edu/news-insights/content/understanding-the-threat-of-harmful-algal-blooms/

When talking about toxin-producing diatoms Pseudo-nitzschia, the similar problems arises in the Adriatic Sea where the results revealed that three investigated species of the genus Pseudo-nitzschia were capable of producing domoic acid indicating their toxic potential. Moreover, toxicological data suggested all three

Pseudo-nitzschia species can excrete toxic secondary metabolites into the surrounding media in addition to the intracellular pools of domoic acid, raising concerns regarding their toxicity and environmental impact.

Smodlaka Tanković et al. (2022) Characterisation and toxicological activity of three different Pseudo-nitzschia species from the northern Adriatic Sea (Croatia). Environ Res. 214(Pt 4): 114108. doi: 10.1016/j.envres.2022.114108.

In 4.1. Strain origin and laboratory acclimation, please indicate in what part of the year did the sampling occur (month)? Better localization of the sampling station would also be advisable (map of the area would be appropriate if possible).

Minor remarks:

Page 5, line 138 – maybe this letter “t” could be removed? (…there was a similar decrease in the compound’s content between the control “t” and the most severe MHW…)

Page 6, line 166 – please revise this sentence to make it a bit clearer “Gymnodinium catenatum is a chain-forming species, and laboratory observations indicate that the longer the chains the better growth conditions the cells are experiencing [15, and references therein]”

Page 8, line 268 – please use mL instead of ml, be consistent in the paper

Author Response

Reviewer #3

The manuscript entitled “Gymnodinium catenatum paralytic shellfish toxin production and photobiological responses under marine heat waves” aimed to investigate how marine heatwaves impact parameters relating to G. catenatum cell growth and toxin production using multi-biomarker approach. Authors concluded that production of paralytic shellfish toxins by G. catenatum does not seem to be potentiated under marine heatwave exposure.

There is a very relevant new document regarding the threat of harmful algal blooms that can be downloaded on the Woods Hole Oceanographic Institution: https://www.whoi.edu/news-insights/content/understanding-the-threat-of-harmful-algal-blooms/

When talking about toxin-producing diatoms Pseudo-nitzschia, the similar problems arises in the Adriatic Sea where the results revealed that three investigated species of the genus Pseudo-nitzschia were capable of producing domoic acid indicating their toxic potential. Moreover, toxicological data suggested all three Pseudo-nitzschia species can excrete toxic secondary metabolites into the surrounding media in addition to the intracellular pools of domoic acid, raising concerns regarding their toxicity and environmental impact.

Smodlaka Tanković et al. (2022) Characterisation and toxicological activity of three different Pseudo-nitzschia species from the northern Adriatic Sea (Croatia). Environ Res. 214(Pt 4): 114108. doi: 10.1016/j.envres.2022.114108.

Reply: We greatly appreciate the interest and the references shared, the first one was incorporated in the text (line 48, reference [9]). It is indeed very interesting that Pseudo-nitzschia is also able to excrete harmful secondary metabolites. It is thought that G. catenatum may produce BMAA (β-N-methylamino-l-alanine) as a secondary metabolite. This compound is associated with neurodegenerative diseases. It would be very interesting to measure the effects of MHWs or any climate-related variable on the production of a suite of secondary metabolites in both HAB species.

  • In 4.1. Strain origin and laboratory acclimation, please indicate in what part of the year did the sampling occur (month)? Better localization of the sampling station would also be advisable (map of the area would be appropriate if possible).

Reply: the information has been added and the text now reads: “The clonal strain of G. catenatum (IO13-04) used in the present study was established by isolation of a single cell from phytoplankton net samples collected in Espinho (41º0'0"N 8º38'56.4"W), western coast of Portugal, in November 2005.” (L. 248-250).

Minor remarks:

  • Page 5, line 138 – maybe this letter “t” could be removed? (…there was a similar decrease in the compound’s content between the control “t” and the most severe MHW…)

Reply: Thank you for noticing. The error has been removed.

  • Page 6, line 166 – please revise this sentence to make it a bit clearer “Gymnodinium catenatum is a chain-forming species, and laboratory observations indicate that the longer the chains the better growth conditions the cells are experiencing [15, and references therein]”

Reply: We understand your comment and changed the sentence to the following: “Gymnodinium catenatum is a chain-forming species, and under laboratory conditions chain formation and length are considered indicators of optimal growth conditions [16 and references therein].” (L. 176-179).

  • Page 8, line 268 – please use mL instead of ml, be consistent in the paper

Reply: Thank you for noticing, we corrected this throughout the manuscript.